# Impact of Systemic Comorbidities on Ocular Hypertension and Open-Angle Glaucoma, in a Population from Spain and Portugal

**DOI:** 10.3390/jcm11195649

**Published:** 2022-09-25

**Authors:** Carolina Garcia-Villanueva, Elena Milla, José M. Bolarin, José J. García-Medina, Javier Cruz-Espinosa, Javier Benítez-del-Castillo, José Salgado-Borges, Francisco J. Hernández-Martínez, Elena Bendala-Tufanisco, Irene Andrés-Blasco, Alex Gallego-Martinez, Vicente C. Zanón-Moreno, María Dolores Pinazo-Durán

**Affiliations:** 1Ophthalmic Research Unit “Santiago Grisolia”, FISABIO (Foundation for Research in Health and Biomedicine), 46017 Valencia, Spain; 2Cellular and Molecular Ophthalmobiology Group, Surgery Department, Faculty of Medicine and Odontology, University of Valencia, 46010 Valencia, Spain; 3Department of Ophthalmology, University General Hospital, 46014 Valencia, Spain; 4Department of Ophthalmology, Clinic Hospital, 08036 Barcelona, Spain; 5Spanish Net of Ophthalmic Pathology OFTARED, Institute of Health Carlos III, 28029 Madrid, Spain; 6Technological Centre of Information and Communication Technologies (CENTIC), 30100 Murcia, Spain; 7Spanish Net of Inflammatory Diseases RICORS, Institute of Health Carlos III, 28029 Madrid, Spain; 8Department of Ophthalmology, The General University Hospital “Morales Meseguer”, Ave/Marques de los Velez s/n, 30008 Murcia, Spain; 9Department of Ophthalmology and Optometry, University of Murcia, 30120 Murcia, Spain; 10Department of Ophthalmology, Punta de Europa Hospital, 11207 Algeciras, Spain; 11Department of Ophthalmology, Jerez Hospital, 11407 Jerez de la Frontera, Spain; 12Department of Ophthalmology, Clinsborges, 4000-422 Porto, Portugal; 13Department of Ophthalmology, San Juan de Dios Hospital, 41930 Sevilla, Spain; 14Department of Biomedical Sciences, Faculty of Health Sciences, University Cardenal Herrera CEU, 46115 Valencia, Spain; 15Faculty of Health Sciences, Valencian International University—VIU, 46002 Valencia, Spain

**Keywords:** systemic comorbidities, ocular hypertension, open-angle glaucoma, risk factors, optic nerve degeneration

## Abstract

Open-angle glaucoma (OAG), the most prevalent clinical type of glaucoma, is still the main cause of irreversible blindness worldwide. OAG is a neurodegenerative illness for which the most important risk factor is elevated intraocular pressure (IOP). Many questions remain unanswered about OAG, such as whether nutritional or toxic habits, other personal characteristics, and/or systemic diseases influence the course of glaucoma. As such, in this study, we performed a multicenter analytical, observational, case–control study of 412 participants of both sexes, aged 40–80 years, that were classified as having ocular hypertension (OHT) or OAG. Our primary endpoint was to investigate the relationship between specific lifestyle habits; anthropometric and endocrine–metabolic, cardiovascular, and respiratory events; and commonly used psychochemicals, with the presence of OHT or OAG in an ophthalmologic population from Spain and Portugal. Demographic, epidemiological, and ocular/systemic clinical data were recorded from all participants. Data were analyzed using the R Statistics v4.1.2 and RStudio v2021.09.1 programs. The mean age was 62 ± 15 years, with 67–80 years old comprising the largest subgroup sample of participants in both study groups. The central corneal thickness (ultrasound pachymetry)-adjusted IOP (Goldman tonometry) in each eye was 20.46 ± 2.35 and 20.1 ± 2.73 mmHg for the OHT individuals, and 15.8 ± 3.83 and 16.94 ± 3.86 mmHg for the OAG patients, with significant differences between groups (both *p* = 0.001). The highest prevalence of the surveyed characteristics in both groups was for overweight/obesity and daily coffee consumption, followed by psychochemical drug intake, migraine, and peripheral vasospasm. Our data show that overweight/obesity, migraine, asthma, and smoking are major risk factors for conversion from OHT to OAG in this Spanish and Portuguese population.

## 1. Introduction

Ocular hypertension (OHT) is the only known modifiable risk factor of glaucoma development. Intraocular pressure (IOP)-lowering therapy reduces the risk of glaucoma. Several risk factors for glaucoma conversion, mainly a higher level of IOP and age, have been established by the Ocular Hypertension Treatment Study (OHTS) and the European Glaucoma Prevention Study (EGPS). However, substantial variability exists in the measurement of the currently known risk factors, especially if the assessment is performed from a longitudinal perspective. Additionally, many factors are responsible for overdiagnosis [1,2,3,4]. Therefore, both under- and overdiagnosis must be eliminated by proper evaluation of all glaucoma suspects (GSs).

A GS is defined as a person who has one or more clinical features and/or risk factors that increase the possibility of developing glaucomatous optic nerve degeneration (OND) and visual deficiency [5,6,7]. The identification of precise glaucomatous pre-perimetric characteristics or biomarkers of the disease is still a pending issue for both ophthalmologists and researchers [5,6,7]. The OHTS and EGPS mainly dealt with the ability of preventive topical hypotensive treatment to decrease the rate of conversion from OHT to OAG [8,9,10,11]. The EGPS documented that topical hypotensive therapy with dorzolamide reduced IOP by 15% to 22% during the five years of follow-up [8,9]. The OHTS concluded that OAG develops within 5 years in 9.5% of untreated OHT individuals and in only 4.4% of treated OHT individuals [10,11].

Open-angle glaucoma (OAG) is the most prevalent type of glaucoma worldwide, and its diagnosis depends on the demonstration of clinical signs such as IOP elevation and optic nerve damage, which can be documented using structural imaging techniques and functional tests [4,5,6,7,9,10,11,12,13].

Recognized non-IOP risk factors for OAG mainly include older age, ethnicity, thinner corneas, and the degree of glaucoma severity [4,5,6,8,9,10,11,13,14]. Additionally, myopia and a family history of glaucoma are important risk factors affecting glaucoma development [8,9,10,11,13,14]. Low ocular perfusion pressure, cardiovascular and/or cerebrovascular disease, systemic hypertension and/or hypotension, diabetes mellitus, and/or hypercholesterolemia are also considered to play important roles in determining the course of OAG [15,16]. Although these and many other researchers have investigated the role of these risk factors in OAG, many only considered the role of IOP changes or were cross-sectional in design, precluding researchers from finding a causal association between a specific risk factor and glaucoma onset. In the meantime, hypotensive medical, laser, and/or surgical treatment are the only options to counteract an elevated IOP [8,9,10,11,12,13,17,18,19,20].

Increased knowledge of the pathological mechanisms and risk factors of OAG is urgently needed. Identifying key non-ocular factors that may influence the progression from OHT to OAG is critical for designing specific interventions to more appropriately managing patients at risk of glaucomatous OND. Because only few studies have been conducted on this topic in Spain and Portugal, a singular geographical area where family and public culture and lifestyle behaviors differ from other European countries [21], we decided to gather information on the conventional and other types of non-ocular risk factors for the conversion of OHT to OAG in an ophthalmological Spanish and Portuguese population. We mainly considered factors involving metabolic, cardiovascular, and respiratory diseases; lifestyle characteristics and habits (body mass index, coffee and/or tea intake, alcohol and/or tobacco use, and psychotropic drug use) that may play an additional role in the development and progression of OAG.

## 2. Materials and Methods

We conducted a multicenter, analytical, observational, case–control study of 635 participants, initially recruited from the ophthalmological departments of the collaborative hospitals from Spain and Portugal, who agreed to participate in the study and signed an informed consent form. We adhered the principles of the Declaration of Helsinki (Edinburgh, 2000) and the Ethics Committee standards of the study centers (2021). The clinical research requirements to maintain the privacy of all data obtained from our study volunteers were met.

Ophthalmologists from the glaucoma sections at the main collaborative university hospitals (General of Valencia; Clinic of Barcelona; Morales Meseguer of Murcia; Punta de Europa of Algeciras; Jerez of Jerez de la Frontera; Clinsborges of Porto; San Juan de Dios of Sevilla and Dr. Peset of Valencia) performed a systematized ocular examination of 635 initial volunteers of both sexes and aged 40 to 80 years, presenting to glaucoma clinics, through a nonrandom consecutive sampling method. Suitable study participants were chosen by ensuring their appropriate status according to the inclusion and exclusion criteria, as listed in Table 1. We divided the final recruited participants into two groups: (1) individuals with OHT (OHTG; *n* = 198) and (2) patients diagnosed with OAG (OAGG; *n* = 214). All patients diagnosed with OAG and some participants of the OHTG were treated with hypotensive eye drops, laser, or glaucoma surgery, depending on the individual needs and glaucoma stage. The major causes of the reduction in sample size from the baseline-selected participants to the end of study were the following: volunteer withdrawal, clinical findings that strongly recommended excluding the participant, personal reasons, or exceptional complications that prevented finishing the study course.

We conducted interviews with each participant to determine the social and demographic factors associated with specific characteristics and lifestyle domains, systemic comorbidities, and commonly prescribed psychochemical medications. All surveyed volunteers answered questions regarding the following: height and weight, which were used to determine the body mass index (BMI) by dividing the weight (kg) by the height (m^2^) (25 to 29.9 indicates overweight and >30 indicates obesity); coffee or tea consumption in the form of hot beverages, recorded as the average number of cups consumed per day or per week or lack of coffee or tea intake; smoking and/or alcohol habits, which was recorded as alcoholic beverages consumed per day or per week and/or how many cigarettes were smoked daily or weekly, with lack of alcohol or tobacco use also recorded; systemic disorders, which included questions about thyroid status (hyperthyroidism or hypothyroidism) and cardiovascular (peripheral vascular dysregulation, and/or migraine) and respiratory (asthma, obstructive chronic pulmonary disease (OCPD), and sleep apnea) conditions as well as the use of psychochemical drugs (prescriptive drugs with potential sedative effect, including those for anxiety, depression, affective or mood disorders, strong painkillers, or sleep medication). Data were recorded as DEMO and QUEST in a Microsoft Excel spreadsheet that was specifically reviewed by the survey supervisors.

Ophthalmic examination was performed by combining the IOP measurements (obtained by Goldman applanation tonometry Haag-Streit AT 900; Haag-Streit Köniz, Switzerland), morphological measurements (ocular fundus exploration with −78 D lens through a slit-lamp; IMAGEnet, Topcon, Barcelona, Spain), and optical coherence tomography (OCT) cirrus spectral-domain OCT (Carl Zeiss Meditec, Inc., Madrid, Spain), as well as functional (visual field (VF)) performance, using the 24-2 Swedish interactive threshold algorithm (Humphrey field analyzer, Carl Zeiss Meditec, Inc., Madrid, Spain) for evaluation. We considered the best-corrected visual acuity (BCVA) obtained by the logarithm of the minimum angle of resolution (LogMAR) for each eye. The IOP was measured three consecutive times during the ophthalmological visit within the time-period between 09:00 a.m. and 12:00 p.m. Only the study professionals were responsible for the IOP measurements by means of the Goldman tonometer. The mean IOP in the normal adult population is 15–16 mmHg. Normal IOP is defined as two standard deviations above the normal, i.e., 21 mmHg; any IOP above this level was considered being elevated. We report IOP values as the mean ± SD in millimeters of mercury for three determinations for each participant.

Central corneal thickness (CCT) was measured using hand-held ultrasonic pachymetry (Reichert^®^ iPac^®^; Reichert/Ametek, Munich, Germany). The pachymeter was calibrated at the beginning of each session following the manufacturer’s recommendations. After instillation of anesthetic eye drops, on an undilated pupil, we performed the ultrasonic pachymetry using a probe tip of approximately 1 mm in diameter. The tip was perpendicularly placed on the plane of the central cornea. Three independent measurements were conducted in a random sequence of 3 min from each other. Normal CCT (ultrasonic pachymetry) was estimated at 533 μm. CCT values are reported as the mean ± SD μm of three determinations for each participant.

IOP and CCT values are well interconnected. The CCT (ultrasound pachymetry)-adjusted IOP (Goldman tonometry) for the study participants was calculated according to the Los Angeles Latino Eye Study [20] for a CCT starting point of 550 μm: Goldmann applanation IOP − (CCT − 550)/50 × [X mmHg]. A diagnosis of OHT was given when the IOP exceeded 21 mmHg, with normal visual fields, normal optic disc, and retinal nerve fiver layer (RNFL) as well as a gonioscopy demonstrating an open anterior chamber angle. OAG is a chronic progressive optic neuropathy with characteristic morphological changes at the optic nerve head (ONH) and RNFL in the absence of other ocular disease or congenital anomalies. Progressive retinal ganglion cell (RGC) death and VF loss are associated with these changes. Glaucoma staging was guided by the Hoddap et al. [22] criteria based on the VF mean deviation (MD) parameter, the quantity of depressed points in the pattern deviation map, and the proximity of VF defects to the fixation point. The defects were classified as either early (MD values of −6 dB or higher), moderate (MD between −6 and −12 dB), or severe (MD of −12 dB or lower) glaucomatous loss. Therefore, glaucomatous damage staging into the appropriate categories was applied to the study participants to facilitate more appropriate management of the disease. Glaucoma OND includes typical optic nerve head alterations such as neuroretinal rim thinning, splinter hemorrhages, peripapillary nerve fiber loss, asymmetry of cupping between patient eyes, and parapapillary atrophy, among others. We recorded the data as OPHTHAL into a Microsoft Excel spreadsheet, which was reviewed by two independent ophthalmologists.

Statistical analyses were conducted by two independent data scientists using R Statistics v4.1.2 and RStudio v2021.09.1, a set of integrated tools designed to help with R (RStudio Team 2021. RStudio: Integrated Development for R. RStudio, PBC, Boston, MA, USA). R packages readxl, dplyr, ggplot2, ggcorrplot, and DemoTools were used for data loading and wrangling, exploratory analysis, and data plotting. R packages caret, MASS, Epi, questionr, and car were used for data modeling and model evaluation. Pearson chi square test was used for comparing two categorical variables, and Shapiro–Wilk test to check the distribution of quantitative variables. Comparison of two means was carried out using Student’s *t*-test for independent samples (normal variables) or the Mann–Whitney U test (non-normal variables). Comparison of more than two means was carried out through analysis of variance (ANOVA, normal variables) or the Kruskal–Wallis test (non-normal variables). Akaike information criterion (AIC) was used to estimate the prediction error (quality of each model) for a given dataset by providing a method for optimizing model election [23,24]. Both eyes of each participant (whenever possible) were included in the statistical processing. Circadian variation in IOP in OAG patients was firmly concordant between both eyes of the same patient. The fellow eye IOP may have asymmetrically fluctuated in short time periods, a factor that was taken to consideration in this study. Collinearity among variables was checked by determining the variance inflation factor (VIF). Values of this factor greater than 5 indicate high collinearity and the variable should be discarded. The significance level was set at 0.05. We recorded data as STAT, which were reviewed by the data scientists.

## 3. Results

### 3.1. Demographic and Epidemiologic Data

The final sample included 412 participants, who we clinically classified as 198 OHT individuals (OHTG) (48.06%) and 214 OAG individuals (OAGG) (51.94%). The distribution of data from each participating hospital was as follows: University Hospital Dr. Peset of Valencia (20%); University General Hospital of Valencia (18%); Clinic Hospital of Barcelona (17%); University Hospital “Morales Meseguer” of Murcia (11%); Punta de Europa Hospital of Algeciras (11%); Clinsborges of Porto, Portugal (11%); University Jerez Hospital of Jerez de la Frontera (6%); and San Juan de Dios Hospital of Sevilla (6%).

The mean age of the study participants was 62.0 ± 15.0 years. The percentages in the different age subgroups among the participants are shown in Figure 1. We found no significant differences between the groups. Most participants were aged 67–80 years in both study groups, as depicted in Figure 1.

Men and women constituted 39.39% and 60.61% (*p* < 0.05) of the OHTG, respectively, versus 51.47% and 48.53% of the OAGG, respectively.

### 3.2. Clinical Characteristics

Because of ophthalmic examination, ophthalmologists assigned volunteers to their respective study groups by integrating the IOP measurements, CCT, ocular fundus exploration findings, OCT parameters, and VF evaluation, all of which were obtained for the right (RE) and left eye (LE) of each participant.

Our data show that mean BC LogMAR VA was 0.00 (RE) and 0.10 (LE) in the OHTG versus 0.20 (RE) and 0.20 (LE) in the OAGG. Moreover, the CCT was significantly higher in the OHTG (RE: 563 ± 9 μm; LE: 571 ± 10 μm) than in the OAGG (RE: 552 ± 10 μm; LE: 548 ± 9 μm). Based on comparison between each eye from both groups, RE: *p* = 0.005; LE: *p* = 0.001. The mean CCT-adjusted IOP in the OHTG was 20.46 ± 2.35 mmHg (RE) and 20.1 ± 2.73 mmHg (LE), whereas in the OAGG, the mean IOP was 15.8 ± 3.83 mmHg (RE) and 16.94 ± 3.86 mmHg (LE). When comparing each eye from both groups, we found significant differences in IOP between the OHTG and OAGG (RE: *p* = 0.001; LE: *p* = 0.001). OCT parameters (mean cup-to-disc ratio, papillary excavation, RNFL thickness, and RGCs density) and VF mean deviation for each eye were significantly different between the study groups (*p* < 0.001).

We categorically considered the relevance of the clinical history to the OHT or OAG diagnosis in this cohort. Age, sex, anthropometric characteristics (BMI), thyroid state, cardiovascular and respiratory situation, smoking and drinking habits, coffee or tea intake, and the use of psychochemical drugs of all participants were determined to help manage the early diagnostic classifications and therapeutic decisions for OHT or OAG individuals. The highest prevalence of the studied characteristics in both groups was for overweight/obesity and daily coffee consumption. Additionally, psychochemical drug intake, migraine, and peripheral vasospasm were outstanding characteristics in this OHT and OAG population, as highlighted in Figure 2.

### 3.3. Statistical Data

First, we performed univariate analysis to determine significant differences between the study groups in some of the variables (Table 2), such as sex (women, *p* = 0.01), IOP (RE/LE, *p* < 0.001), smoking (*p* < 0.001), height (*p* = 0.01), and COPD (*p* < 0.001).

Second, we analyzed the correlation between the studied variables to determine the power of the linear relationship between two variables as well as to record their association in relation to the primary study endpoint, as shown in Figure 3. The “age group” variable was qualitative-ordinal, so the linear (L), quadratic (Q), cubic (C), and fourth power (^4) trends were analyzed.

Third, we analyzed the diagnostic variable using logistic regression (based on the other 16 variables, as listed in Table 2). The results of the analysis are shown in Table 3. The following four variables were found to significantly influence the diagnosis (OHTG vs. OAGG): BMI (*p* = 0.03), asthma (*p* = 0.003), COPD (*p* = 0.03), and migraine (*p* = 0.001).

Therefore, we repeated the logistic regression analysis, performing an iterative elimination of variables using the AIC. Now, we found that three of the four previous variables (as reflected in Table 3) maintained their statistically significant influence on the diagnosis (OHTG vs. OAGG): BMI (*p* = 0.001), asthma (*p* = 0.002), and migraine (*p* = 0.001). In addition, a fourth variable had a significant influence on the diagnosis: weight (*p* = 0.004; Table 4).

Finally, we consider the possible influence of the variables age and gender on the diagnosis of HTO/OAG. Therefore, a third analytical model was created by adding these two variables to the ones pertaining to our previous model, as listed in Table 4. Before this analysis, we checked the independence of the COPD, age, and smoking variables, and we found a significant association between smoking and COPD (*p* < 0.001). These data may be interpreted as COPD being caused by cigarette smoking, which evidently is the main cause of the disease, and is thought to account for around 9 in every 10 cases worldwide. Thus, we decided to remove the variable COPD from the third analytical model, being now, and according to age and gender, the most significant characteristics in our study population: smoking (*p* = 0.041), weight (*p* = 0.009), BMI (*p* = 0.001); asthma (*p* = 0.002), COPD (*p* = 0.044), and migraine (*p* = 0.002). Overall, these data are reflected in Table 5.

## 4. Discussion

We explored some classical and emergent risk factors in HTO or OAG men and women volunteers pertaining to a Spanish and Portuguese ophthalmological population aged 40 to 80 years, revealing the importance of identifying people more predisposed to progression from OHT into OAG, as well as to early undertaking precision measures for avoiding OND and visual impairment-blindness [4,5,6,7,8,9,10,11,12,13,25].

In this concern, four comorbidities significantly influenced the diagnosis of OHT vs. OAG, being these: overweight/obesity, migraine, asthma, and smoking.

OAG is a chronic optic neuropathy following the damage and death of the RGCs in the framework of IOP elevation, and it remains the leading cause of irreversible blindness worldwide [25,26,27]. However, the mechanisms of OAG pathology are not fully understood.

Individuals with OHT constitute an important group to manage because they are frequently undiagnosed. Additionally, GSs are at high risk of OND with respect to the global population [5,6,7,28]. Standard automated perimetry-based VF performance remains the benchmark for assessing the intensity of glaucomatous injury [5,6,7,8,9,10,11,12,29,30]. Epidemiological and experimental studies have shown glaucomatous RGC loss precedes the apparition of VF defects [29,30,31,32,33,34] because structural alterations occur before the onset of functional changes [35,36]. Therefore, the absence of symptoms and inadvertent signs, mainly during the earlier glaucoma stages (such as in pre-perimetric glaucoma cases and early perimetric glaucoma patients), complicates glaucoma diagnosis and overshadows the visual prognosis of affected individuals [5,6,7,32,33,34,35,36]. Closely monitoring OHT individuals, GSs, and OAG patients minimizes the risk of unidentified disease progression [27,29].

In this study, most participants were in the age range of 67–80 years in the OHTG and OAGG (Figure 1). Several authors [4,15,16,28,37] found that OAG is conventionally associated with other ocular (cataracts and macular degeneration) and systemic (vascular, respiratory, metabolic, and neurologic) age-related diseases. Frailty is defined as the state of being frail, with interest in the frail elderly growing worldwide, as they are especially susceptible to develop disabilities [37,38]. According to McMonnies [28], Fulop et al. [38], and Rockwood et al. [39], the frailty of health and disability in the elderly depends on the accumulation of deficits throughout the adult lifespan. In this respect, patients with OAG are expected to suffer other health and wellbeing problems (hypertension blood pressure, type 2 diabetes mellitus, etc.), which in turn may impact glaucoma development and course [28,37,40], which is our primary endpoint in this study.

Exclusive reliance on clinical glaucoma tests may decrease the likelihood of identifying OHT subjects at higher risk of progression to glaucoma. Moreover, total reliance on factual data regarding glaucoma hallmarks may negatively influence the decision to treat individuals. Several authors [15,16,27,28,40] have emphasized the importance of systemic diseases in glaucoma diagnosis and therapy management. Regardless of age, race, family history, and thinner cornea, from an epidemiological viewpoint, hypertension blood pressure, diabetes mellitus, and obesity are correlated with glaucoma [4,5,6,13,14,15,16,25,28,40,41]. Thus, chronic systemic diseases and/or age-related disorders may act as risk factors for the progression from OHT to OAG as well as OAG progression [15,16,27,28,37,38,40,41]. Overall, information may be improved by specific clinical history findings related to lifestyle features, anthropometry, toxic habits, medication, and diverse comorbidities, including endocrine–metabolic, cardiovascular, and respiratory pathologies. In our study population, we evaluated the following comorbidities and lifestyle characteristics: overweight/obesity, thyroid dysfunction, habitual coffee and tea consumption, alcohol and smoking habits, psychochemical drug intake, peripheral vasospasm, migraine, asthma, COPD, and sleep apnea. The most common characteristics in both study groups were overweight/obesity, daily coffee consumption, psychochemical drug intake, migraine, and peripheral vasospasm (Figure 2). The analyzed comorbidities and lifestyle characteristics are discussed below.

**Overweight and obesity** are conditions in which excessive fat storage poses a serious risk to health and wellbeing [1,2,3,4,8,9,10,11,12,13,14,15,27,42,43], and they have reached global epidemic status and are a major risk factor for serious diseases, including those affecting the eyes and vision [15,16,25,37]. A prospective population-based cohort from the Rotterdam study [44] demonstrated that 2.7% of people developed OAG during a 10-year follow-up, with women with a high BMI having a significantly higher chance of developing OHT but lower risk of developing OAG. In contrast, a two-database matched-cohort study conducted in Taiwan strongly suggested that obese adults display an elevated OAG risk [45]. Jung et al. [46], in a nationwide Korean population-based study, drew the same conclusion that metabolic health status and obesity are significantly associated with augmented OAG risk. Likewise, Lin et al. [47] described a causal association between obesity and OAG in a two-sample Mendelian randomized study performed in China. In our prospective population study, half of the OHT individuals and the OAG patients were overweight/obese. Furthermore, statistical processing revealed that BMI significantly influenced the diagnosis of HTO and OAG, according to age and sex.

**Coffee and tea** are the hot drinks most consumed worldwide, with consumption significantly increasing from 2012 to 2021 [48]. The medicinal properties of these hot drinks have been acknowledged for centuries, which primarily come from the effects of caffeine as an antioxidant, anti-inflammatory, and antiapoptotic agent. However, its widespread use as a stimulant has induced a myriad of studies with contradictory reports regarding the effects of caffeine on health [49]. Tellone and Galtieri [50] reported that caffeine exerts protective effects against neurodegenerative disorders. However, higher caffeine levels also provoke hypercholesterolemia, insomnia, cerebral stroke, and myocardial infarction, among others [51]. Habitual coffee consumption in relation to OAG has been extensively evaluated [1,2,3,4,8,9,10,11,12,13,14,15,28]. A prospective cohort study conducted by Kang et al. [52] first confirmed that caffeine intake is not associated with increased OAG risk, but a secondary data analysis showed that caffeine augments the risk of OAG in the participants with a family history of glaucoma. In a study conducted in China by Li et al. [53], single-nucleotide polymorphisms (SNPs) associated with coffee consumption (phenotypes 1 and 2) were found in a genome-wide association study (GWAS) of the study population of people of European ancestry, confirming that higher coffee consumption is associated with a higher OAG risk. Regarding tea consumption, Wu et al. [54] found that participants who consumed hot tea daily were less likely to have glaucoma than non-consumers. Additionally, green tea has been considered by several authors as a potential neuroprotective agent due to its antioxidant properties [55]. In our study population, the studied characteristics with the highest prevalence were overweight/obesity and daily coffee consumption. However, we found no correlation between the consumption of these two hot drinks and OHT/OAG risk. Nevertheless, controversy exists regarding the benefits and risks of these substances on eye and vision health.

**Psychochemicals** are widely prescribed for the treatment of a variety of mental illnesses (depression, anxiety, affective and mood disorders, psychosis, epilepsy, and insomnia). Drugs of the new generation are currently the first line treatment for affected patients, among which includes the selective serotonin reuptake inhibitors (SSRIs) and serotonin noradrenaline reuptake inhibitors (SNRIs), among others [56]. These drugs produce substantial side effects, most of them being transient, but others are potentially severe and may continue even after cessation of drug treatment. Additionally, strong evidence supports the increase in acute and chronic mental health disorders related to the SARS-CoV-2 pandemic, for which specific care is required [57]. Psychochemical drugs have been frequently associated with glaucoma [1,2,3,4,8,9,10,11,12,13,14,15,28]. For example, gabapentin and carbamazepine induce anterior eye chamber deepening, iridocorneal angle widening, and mild pupillary dilation. Moreover, carbamazepine is associated with significant increase in IOP independent of gabapentin intake [58]. However, Wang et al. [59], in a systematic review and meta-analysis investigating the relationship between antidepressant drug consumption and OAG, reported that serotoninergic antidepressants did not represent a higher risk of glaucoma. In our study population, 36% of the OHTG and 37% of the OAG patients suffered psychologic, psychiatric, and/or sleep disorders, having been prescribed the corresponding treatment. Regardless, we found no statistical significance link between these treatments and OHT/OAG risk.

**Alcohol and tobacco** are considered lifestyle factors and sociocultural habits as well as the most addictive legal drugs with the highest popularity of consumption in the general population, with both having an important impact on morbidity and **mortality worldwide** [60]. Polydrug use and/or abuse, as in cases of joint alcohol and tobacco habits, have even more detrimental consequences for health, life, and families. Serious consequences in terms of absences from work among employees who both smoked and drank at least weekly have also been reported [60]. Several population-based epidemiological studies did not find any association between alcohol consumption and OAG risk [28,61]. We found no significant differences between groups in this study population regarding alcohol consumption. Perez de Arcelus et al. [62] found a direct association between current smokers and the incidence of glaucoma. Zanón-Moreno et al. [63] analyzed aqueous humor and plasma samples from smokers, nonsmokers, and ex-smokers among elderly glaucomatous women, revealing that expression levels of interleukin-6, caspase-3, and poly (ADP-ribose) polymerase 1 (PARP-1) were significantly increased in smokers, indicating that smoking is an important additional risk factor for progression of OAG. In this study, smokers accounted for 29% of the OHTG and 14% of the OAGG, and drinkers constituted 18% of the OHTG and 17% of the OAGG. The results of univariate analyses showed significant differences between the OHTG and OAGG in the variables sex (women, *p* = 0.01) and smoking (*p* < 0.001), similarly to the findings by Zanon-Moreno et al. [63].

**Comorbidities** are common in glaucoma [15,16,28,37,44,45,46,47]. We analyzed endocrine–metabolic, cardiovascular, and respiratory diseases in this study. From a practical viewpoint, we focused on thyroid gland dysfunction, migraine, peripheral vasospasm, asthma, COPD, and sleep apnea; the results are discussed below. The **thyroid gland** is located on the anterior cervical area and produces thyroid hormone, which plays a pivotal role in controlling general metabolism, immune system, mood state and cognitive functions, and nearly every system in the body [64]. Dysfunction of the thyroid gland affects the proper management of the cardiovascular, endocrine–metabolic, immune, neuromuscular, gastrointestinal, reproductive, and adrenal systems, among numerous others [65,66]. The first reports linking the thyroid status and OAG were published by McLehachan and Davies [66]. In a population-based study of the elderly participants in the Blue Mountains Eye Study [67], thyroid dysfunction was found to be independently related to OAG. Other authors have mainly considered the relationship of thyroid-associated orbitopathy and glaucoma [68]. In our cohort, hypothyroidism or hyperthyroidism was present in 19% of the OHTG and 14% of the OAGG. The relationship between the OHT and OAG was not statistically significant after adjusting for potential confounders in this population. Vascular mediators and hypoxia may be involved in the etiology of eye disorders, including glaucoma.

Because of this, we also formulated some questions to obtain a global view of the trends related to vascular disorders in this population cohort. **Migraine** is a devastating condition that ranks among the most disabling disorders worldwide and has been considered an important OAG risk factor, with studies yielding controversial conclusions [69,70]. A meta-analysis by Xu et al. [71] suggested that migraine can significantly elevate the risk of OAG, but the cohort study design failed to identify this association. Huang et al., [72] found an association of migraine with a higher OAG risk in patients aged <50 years. Other authors have described the relationship between OAG and vascular risk factors [15,16,28,37]. In our population, 31% of the OHT individuals and 18% of the OAG patients suffered migraines. From the results of logistic regression analysis, we found that experiencing migraines significantly influences the diagnosis of OHT vs. OAG. **Peripheral vasospasm** is vascular dysregulation (altered vasoconstriction and/or vasodilation) of the microvasculature that is manifests as symptoms of cold hands and feet [73] and is associated with glaucoma risk and progression [74,75]. We identified this symptomatology in 25% of our OHTG and 31% of our OAGG populations. We observed no other significant differences between groups in this study population for this disorder in relation to OHT and OAG.

Regarding the respiratory disorders that we analyzed in our study population (asthma, COPD, and sleep apnea), our data are described below. **Asthma** is one of the most frequent respiratory illnesses globally, affecting both sexes of all ages [76]. Hallmarks of the disease are dyspnea, wheeze, cough, and chest tightness. Asthma patients require high doses of inhaled, oral, or parenteral steroids as well as other treatments to control the disease [77,78]. **COPD** is a multifactorial disease that includes emphysema and chronic bronchitis, associated with pulmonary symptoms (increasing breathlessness, persistent cough, and wheezing, as well as constant chest infections) and a variety of extrapulmonary manifestations. Smoking is the major risk factor for COPD. Short-acting bronchodilator inhalers are the first line therapy for the disease [78,79]. Occasionally, distinguishing between asthma, COPD, and sleep apnea complications and its drug-related symptomatology, as well as true primary comorbidities, is challenging. Glaucoma and cataract may be assigned as side effects of asthma and COPD treatment [79]. Moreover, asthma and COPD are associated with numerous comorbidities, the latter to a greater extent. Jassim et al. [80] reported that oxidative stress and hypoxia strongly modify the mitochondrial homeostasis in a mice chronic glaucoma model. **Sleep disordered breathing** includes a wide spectrum of anomalies, including sleep apnea, with a common feature of hypoventilation–hypoxia, playing an important role in the pathogenesis of cardiovascular, neurologic, and metabolic disorders [81]. Cesareo et al. [82] reported that the hypoxia created by the apnea/hypopnea cycles may be a major cause of vascular and neurological dysfunction in terms of the pathogenic mechanisms of OAG. In our cohort study, the prevalence of respiratory disorders in the OHTG and OAGG was 9% and 12% for asthma, 16% and 5% for COPD, and 13% and 11% for sleep apnea, respectively. By logistic regression, asthma was shown to significantly influence the diagnosis of OHT vs. OAG, according to age and gender. COPD and sleep apnea did not show any statistical difference between the groups. More studies are needed on these respiratory pathologies and their relationship with OAG.

Overall, our data show that overweight/obesity, migraine, asthma and smoking, are major risk factors for the progression from OHT to OAG in our study population.

The limitations of the study are as follows: (1) Participants may have underreported their characteristics, lifestyle, and illnesses, either intentionally or due to recall bias. (2) Volunteers may have underreported their own tobacco and alcohol use habits. Moreover, binge drinkers (who are less likely to provide correct answers reflecting actual events in surveys) may have provided inaccurate answers our questions. (3) Although we could not accurately categorize systemic disorder for all cases, we considered the most relevant issues for each comorbidity in the study endpoints. (4) We did not include the medications for each systemic disease surveyed or for glaucoma. (5) The study produced a large amount of information, and the statistical processing created large volumes of data. Due to this, we focused on the main study objectives, and some of the information and data have been omitted from the final data in processing. Briefly, we took some actions to mitigate the above limitations, such as by confirming the patient data collection through an extensive review of their clinical charts and by speaking with the accompanying persons and/or family members. Moreover, we discussed any discrepancy in the selection of suitable participants, data screening, and results. We are aware of that any of these described potential limitations could have affected the study findings and interpretations. However, to ensure coherence in the obtained information, data scrubbing and normalization were independently performed by two researchers. All data reported herein will be deposited in a cross-disciplinary public repository. These actions represent a logical and useful way of improving our data power.

Holistic approaches to glaucoma must be taken by selecting the proper strategies that permit ophthalmologists to address the whole person rather than only evaluating the ocular status. The patient account of their medical history is a fundamental step in any medical act. In the era of precision medicine, glaucoma specialists must properly evaluate the risk factors related to lifestyle, non-ocular-specific personal characteristics, and systemic comorbidities. The data from our study consistently indicate that individuals with OHT and OAG suffering from overweight/obesity, migraine, or asthma, and the smokers are at higher risk of progression from OHT to OAG and glaucoma. Through extensive control and/or modification of the above disorders, the development or progression of glaucoma can be positively altered in the affected persons.

## 5. Conclusions

In this Spanish and Portuguese cohort, we identified some non-ocular-specific factors to help assess the risk of OAG and to prevent or delay glaucoma OND and blindness, namely overweight/obesity, migraine, asthma, and smoking. Based on the results of this study, we strongly recommend careful and considered care for the people suffering from these disorders to prevent subsequent visual disability and loss of quality of life.

## Figures and Tables

**Figure 1 jcm-11-05649-f001:**
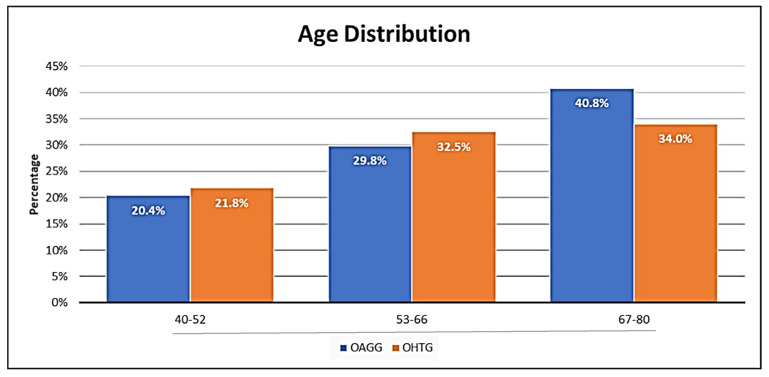
Study participants according to age subgroups, from 40 to 80 years of age. OAGG: open-angle glaucoma group; OHTG: ocular hypertension group.

**Figure 2 jcm-11-05649-f002:**
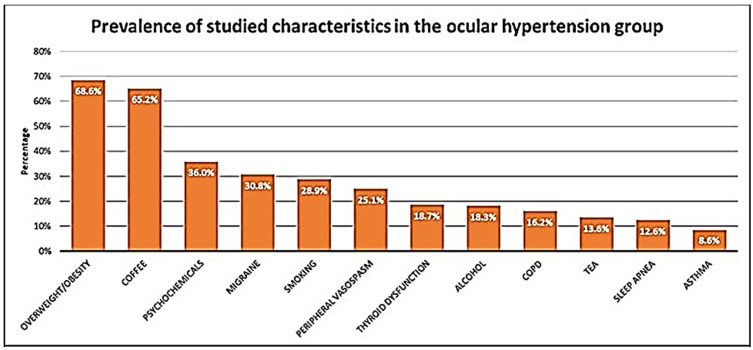
Prevalence of the characteristics in the study population (OHTG and OAGG). COPD: chronic obstructive pulmonary disease.

**Figure 3 jcm-11-05649-f003:**
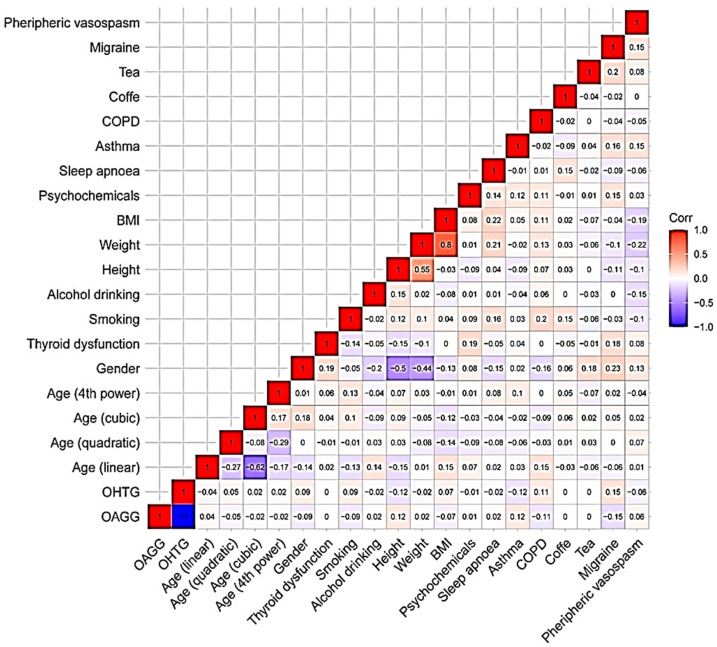
Correlation analysis between the studied variables. COPD: chronic obstructive pulmonary disease; BMI: body mass index; OHTG: Ocular hypertension group; OAGG: Open-angle glaucoma group.

**Table 1 jcm-11-05649-t001:** Inclusion and exclusion criteria for the study participants.

**Inclusion Criteria**	
**OHTG**	**OAGG**
Diagnosis of OHT without glaucoma OND signs	Diagnosis of OAG (early or moderate glaucoma stage)
Age 40–80 years	
Both sexes	
Capacity to understand and participate in the study	
**Exclusion Criteria**	
**OHTG**	**OAGG**
Glaucoma OND signs	Glaucoma type other than OAG
Other confounding ONH signs	Advanced glaucoma stage
<40 or >80 years old	
Other ocular diseases or systemic pathologies that may have interfered with study objectives	
Other treatments, ocular surgery, or laser treatment during last three months that may have interfered with study results	
Unable to participate in study	

OHTG: Ocular hypertension group; OAG: open-angle glaucoma group; OND: optic nerve degeneration; ONH: optic nerve head.

**Table 2 jcm-11-05649-t002:** Results of univariate analysis.

Variable		OAGG	OHTG	*p* *
Diagnostic		214 (51.9)	198 (48.1)	-
Age		62.5	60.8	0.665
Women		108 (51.2)	77 (39.1)	**0.019**
IOP (mmHg)	RE	15.8 ± 3.8	20.5 ± 2.3	**<0.001**
	LE	16.9 ± 3.9	20.1 ± 2.7	**<0.001**
Thyroid dysfunction		30 (14)	37 (18.7)	0.254
Smoking		30 (14)	56 (28.3)	** <0.001 **
Alcohol drinking		37 (17.3)	36 (18.3)	0.895
Height (cm)		165 ± 10	163 ± 9	**0.012**
Weight (kg)		72.8 ± 14.6	72.4 ± 12.9	0.774
BMI (kg/m^2^)		26.5 ± 4.5	27.3 ± 4.5	0.095
Psychochemicals		78 (36.4)	74 (37.4)	0.926
Sleep apnea		23 (10.7)	25 (12.6)	0.660
Asthma		26 (12.1)	17 (8.6)	0.307
COPD		11 (5.1)	32 (16.2)	**<0.001**
Coffee		137 (64.0)	129 (65.1)	0.891
Tea		23 (10.7)	27 (13.6)	0.456
Migraine		38 (17.8)	61 (30.8)	**0.003**
Peripheral vasospasm		67 (31.5)	48 (25.1)	0.195

Data are shown as mean ± standard deviation for quantitative variables or *n* (%) for qualitative variables. OAGG: open-angle glaucoma group; OHTG: ocular hypertension group; IOP: intraocular pressure; RE: right eye; LE: left eye; BMI: body mass index; COPD: chronic obstructive pulmonary disease. Student’s *t*-test for independent variables was used to compare 2 quantitative variables. Chi square test was used to compare two qualitative variables. * Significance at 0.05. The row for each statistically significant variable is shown with a dark background.

**Table 3 jcm-11-05649-t003:** Results of logistic regression analysis using all study variables.

Variable	Coefficient	SD	Z-Value	*p* *	OR
Intercept	−10.188	7.835	−1.300	0.193	-
Age (lineal)	−0.320	0.532	−0.603	0.546	0.726
Age (quadratic)	0.571	0.449	1.271	0.203	1.771
Age (cubic)	0.023	0.319	0.075	0.940	1.024
Age (4th power)	0.196	0.234	0.839	0.401	1.217
Sex	−0.050	0.307	−0.165	0.869	0.951
Thyroid dysfunction	0.252	0.350	0.719	0.472	0.777
Smoking	−0.501	0.329	−1.524	0.127	1.651
Alcohol drinking	0.017	0.326	0.054	0.957	0.983
Height (cm)	5.601	4.825	1.161	0.245	270.816
Weight (kg)	−0.115	0.060	−1.923	0.054	0.891
BMI (kg/m^2^)	0.339	0.157	2.157	**0.031**	1.405
Psychochemicals	0.208	0.252	0.828	0.407	0.812
Sleep apnea	0.216	0.421	0.514	0.607	0.805
Asthma	1.365	0.465	2.934	**0.003**	0.255
COPD	−0.938	0.446	−2.103	**0.035**	2.555
Coffee	0.103	0.245	0.420	0.674	0.902
Tea	0.139	0.397	0.350	0.725	0.870
Migraine	−0.946	0.293	−3.226	**0.001**	2.577
Peripheric vasospasm	0.225	0.269	0.837	0.402	0.798

SD: standard deviation; OR: odds ratio; BMI: body mass index; COPD: chronic obstructive pulmonary disease. * Significance: 0.05. The row for each statistically significant variable is shown with a dark background.

**Table 4 jcm-11-05649-t004:** Results of logistic regression analysis with Akaike information criterion for iterative elimination of variables.

Variable	Coefficient	SD	Z-Value	*p* *	OR
Intercept	−1.328	0.693	−1.917	0.055	-
Smoking	−0.557	0.305	1.822	0.069	1.745
Weight	−0.041	0.014	−2.869	**0.004**	0.959
BMI	0.139	0.044	3.140	**0.001**	1.149
Asthma	−1.382	0.450	−3.067	**0.002**	0.251
COPD	0.797	0.428	1.860	0.063	2.219
Migraine	0.864	0.272	3.173	**0.001**	2.372

SD: standard deviation; OR: odds ratio; BMI: body mass index; COPD: chronic obstructive pulmonary disease. * Significance at 0.05. The row for each statistically significant variable is shown with a dark background.

**Table 5 jcm-11-05649-t005:** Logistic regression analysis with “Akaike Information Criterion” for iterative elimination of study variables, by adding AGE and GENDER.

Variable	Coefficient	SD	Z-Value	*p* *	OR	95% CI	GVIF	Df	GVIF^1/(2*Df)^
**Intercept**	−1.341	0.751	−1.785	0.074	-				
**Age (lineal)**	−0.148	0.502	−0.296	0.767	0.862	0.317–2.336	1.241785	4	1.027438
**Age (quadratic)**	0.528	0.432	1.221	0.222	1.696	0.720–4.011			
**Age (cubic)**	0.051	0.311	0.165	0.869	1.053	0.569–1.942			
**Age (4th power)**	0.171	0.229	0.750	0.453	1.187	0.759–1.860			
**Gender**	−0.103	0.288	−0.356	0.727	0.902	0.510–1.584	1.651071	1	1.284940
**Smoking**	0.624	0.305	2.049	**0.041**	1.866	1.029–3.410	1.062069	1	1.030567
**Weight**	−0.046	0.017	−2.594	**0.009**	0.955	0.922–0.988	4.825147	1	2.196622
**BMI**	0.162	0.051	3.188	**0.001**	1.175	1.066–1.301	4.092805	1	2.023068
**Asthma**	−1.399	0.453	−3.103	**0.002**	0.247	0.096–0.572	1.079335	1	1.038910
**Migraine**	0.849	0.278	3.062	**0.002**	2.338	1.364–4.059	1.099779	1	1.048704

SD: standard deviation; OR: odds ratio; CI: confident interval; Df: degrees of freedom; BMI: body mass index; GVIF: generalized variance inflation factor; GVIF1/(2*Df): standardized GVIF. * Significance was set at 0.05. The row for each statistically significant variable is shown with a dark background.

## Data Availability

Data from this study are contained within this article. All the data showed herein will be deposited in a cross-disciplinary public repository.

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
