# Peer review of "Impact of Systemic Comorbidities on Ocular Hypertension and Open-Angle Glaucoma, in a Population from Spain and Portugal"

_jcm, 2022, doi:10.3390/jcm11195649_

Round 1

Reviewer 1 Report

The authors tried to find the influences of systemic comorbidities on ocular hypertension and open-angle glaucoma, which would be a useful index for potential glaucoma patient management. Here are my comments:

1.     The author mentioned that the purpose of this study was to identify non-ocular specific personal characteristics, systemic comorbidities, and lifestyle facts that may be involved in the interface between OHT and OAG. As previous studies had proofed that abundant risk factors may potentially affect ocular pressure among OHT and OAG  patients (mentioned in the Introduction section, paragraph 3), what is the main innovation point of the current study when compared to the previous study?

2.     The introduction section is too long to get the essence of the core content, this section may need to be shortened and reorganized. The reason of the how OHT /OAG patients’ IOP had been studied and why this study was performed should be carefully addressed.

3.     In method section, definitions of OHT, OAG should be mentioned. For example, diagnosing standard of OND sign, grading standard of OHT, diagnosing standard of mild glaucoma stage, and so on.

4.     In Methods section, the authors mainly collected 635 patients and parts of them suffered from OAG, did these patients had ever been treated either by medicine or surgery? Why had they not been treated or how did the author manage the influence of medicine on IOP?

5.     Abbreviations should be defined carefully when they first appeared. For example, DEMP, QUEST, OPHTHAL, STAT. Please check thoroughly.

6.     Please specify the main packages used in R. Also, please supply Python details as well as employed packages.

7.     According to the methods, the authors mainly tried to perform a multi-center study. The distribution of data from each center should be detailed in the results section, which would address the data bias concerns.

8.     Y-axis as well as Y-scale should be shown in Figure 1.

9.     BCVA data should be treated after LogMAR transformation.

10.   The author should detail how CCT was measured (distance between which planes).

11.   As IOP data would be fluctuant even in continuous measuring, the author should mention how IOP data was collected (how many times were measured by how many physicians, and how the data were finally recorded).

12.   What does the bold margin mean in Figures? Details should be mentioned in the figure legend.

13.   In addition to Q12, what does the exact means in lines with dark background?

14.   What does the exact meaning of legends in Y-axis. For example, Coffee1, COPD1, Alcohol1 and so on.

15.   Since OAG is usually a bilateral disease, if the IOP was significantly different between two eyes, the author should mention why IOP were different. Otherwise, the data should be combined and treated as a whole.

16.   Before logistic regression analysis using all study variables, did the author ever check the collinearity among variables? Results should be detailed shown.

17.   Did the author ever check the cross-function among variables?

18.   95% CI of OR should be supplied in Table 4.

19.   In last paragraph of results section, “HTO/OAG” should be “OHT/OAG”?

20.   In the discussion section, since the author mentioned that tea consumption may be a potential risk factor in affecting IOP, positive or negative relations between tea consumption or tea related consumption dose should be mentioned. Because throwing out only a concept would be useless.

21.   The discussion is too long to extract a core. The author needs to reorganize the discussion section and highlight the key results.

22.   Why did the author mention thyroid gland after alcohol and tobacco part? Reorganizing the discussion was highly recommended.

23.   In the conclusion section, the author should highlight the key findings and meaning of affecting IOP. In the current version, the author only threw out a concept without clear indicators. For example, tea consumption and coffee consumption, the glaucomatous suspects should higher or decrease the frequency and consumption.

Author Response

ANSWERS TO THE REVIEWERS

The Reviewers have brought up some points on our work, and we really appreciate the time used by the Guest Editor and the Reviewers and the opportunity to clarify our research. We are extremely thankful for the Reviewers constructive comments and suggestions that helped our writing skills.

REVIEWER 1

Open Review

English language and style (x) Moderate English changes required  

WE HAVE SENT THE MS TO ENGLISH EDITING. The new version has been received today and it was sent to MDPI submitting system.

We have addressed extensive changes in the English language and style throught the ms and illustrations. Anyway, we asked MDPI for a language review of this article in order to much improve the final version after being published.

Comments and Suggestions for Authors

The authors tried to find the influences of systemic comorbidities on ocular hypertension and open-angle glaucoma, which would be a useful index for potential glaucoma patient management.

Here are my comments:

  1. The author mentioned that the purpose of this study was to identify non-ocular specific personal characteristics, systemic comorbidities, and lifestyle facts that may be involved in the interface between OHT and OAG. As previous studies had proofed that abundant risk factors may potentially affect ocular pressure among OHT and OAG patients (mentioned in the Introduction section, paragraph 3), what is the main innovation point of the current study when compared to the previous studies?

We thank the Reviewer for calling our attention to this important point.

In agreement with the Reviewer, we would like to add some comments. Different studies have previously analyzed the role of a variety of factors in the conversion of OHT to OAG, mainly focusing on the elevated and sustained IOP. In fact, various multi-center randomized studies have shown that the increased IOP is a pivotal risk factor for the development of OHT, as well as for the OAG onset and progression. Recognized non-IOP risk factors are mainly older age, thinner CCT, and the degree of severity of glaucoma. Also, myopia and the family history of glaucoma has been considered important risk factors affecting glaucoma. It has also been described that low ocular perfusion pressure, cardiovascular/cerebrovascular disease, systemic hypertension/ hypotension, diabetes mellitus and/or hypercholesterolemia may play important roles in OAG course. We consider the necessity of  identifying other non-ocular risk factors for the conversion of OHT to OAG, mainly those involving systemic disorders and lifestyle, with the main goal of assessing the relationship between metabolic, cardiovascular and respiratory diseases, lifestyle facts and habits (body mass index, coffee/tea intake, alcohol/tobacco habits, psychotropic drug use) that could play an additional role in the development and progression of OAG, with outstanding therapeutical implications. Moreover, scarce studies on this subject have been done Spain and Portugal, a singular geographical area where the family and public culture, and lifestyle behaviours differs from other European countries.

Moreno-Montañés J, Gándara E, Gutierrez-Ruiz I, Moreno-Galarraga L, Ruiz-Canela M, Bes-Rastrollo M, Martínez-González MÁ, Fernandez-Montero A. Healthy Lifestyle Score and Incidence of Glaucoma: The Sun Project. Nutrients. 2022;14(4):779.

In this sense, our study deals with a remarkable number of systemic factors that may have an impact on glaucoma.

New sentences and two more references (21 and 22) have been introduced (marked in yellow) at the end of the new INTRODUCTION and BIBLIOGRAPHY sections, regarding the Reviewer comments and suggestions. Many thanks, again, to the Reviewer for outstanding help to improve our work.

  1. The introduction section is too long to get the essence of the core content, this section may need to be shortened and reorganized. The reason of the how OHT /OAG patients’ IOP had been studied and why this study was performed should be carefully addressed.

We extremely thank the Reviewer for taking the necessary time and effort to review our work. The introduction has been completely restructured and shortened (marked in yellow), and two more references have been added, according to the Reviewer comments and suggestions regarding this section, and the whole text is enclosed below.

Ocular hypertension (OHT) is the only known modifiable risk factor of glaucoma development. Intraocular pressure (IOP)-lowering therapy reduces the risk of glaucoma development. Several risk factors for glaucoma conversion mainly age and a higher level of IOP were established by the Ocular Hypertension Treatment Study (OHTS) and the European Glaucoma Prevention Study (EGPS). However, there is significant variability in the measurement of the currently known risk factors, especially if the assessment is taken from a longitudinal perspective. Besides, an important number of factors are responsible for over-diagnosis (1-4). It is therefore essential to eliminate both under- and over-diagnosis by proper evaluation of all glaucoma suspects (GS).

A GS is defined as a person who has one or more clinical features and/or risk factors which increase the possibility of developing glaucomatous optic nerve degeneration (OND) and visual deficiency in the future [5,6]. The identification of precise glaucomatous pre-perimetric characteristics or biomarkers of the disease is still a pending issue for both ophthalmologists and researchers [7]. The OHTS and EGPS dealt mainly with the ability of preventive topical hypotensive treatment to decrease the conversion from OHT to OAG. The EGPS documented that topical hypotensive therapy with dorzolamide reduced IOP by 15% to 22% during the five years follow-up [10]. The OHTS concluded that OAG developed within 5 years in 9,5% of not treated OHT individuals, and only in 4,4% of treated OHT individuals [11].

Open-angle glaucoma (OAG) is the most prevalent glaucoma type worldwide, and its diagnosis depends on the demonstration of clinical signs such as IOP elevation and optic nerve damage, that can be documented using structural imaging techniques and functional tests [4-6,9-13, 17].

Recognized non-IOP risk factors for OAG are mainly older age, ethnicity, thinner corneas, and the degree of severity of glaucoma [4-6, 8-10, 13, 14]. Also, myopia and the family history of glaucoma has been considered important risk factors affecting glaucoma [8-10, 13, 14]. In addition, low ocular perfusion pressure, cardiovascular/cerebrovascular disease, systemic hypertension/ hypotension, diabetes mellitus and/or hypercholesterolemia have described to play important roles in OAG course [15, 16].  Although these and many other studies have investigated the role of these risk factors in OAG, many of them have only considered the role of the IOP changes, and many studies were cross-sectional in design and unable to find a causal association between a specific risk factor and the glaucoma onset

Growing awareness on the pathological mechanisms and risk factors of OAG is urgently needed. In the meantime, hypotensive medical, laser and/or surgical treatment are the only choice to counteract the elevated IOP [8-13, 18-21]. Identifying non-ocular key factors that may influence the conversion from OHT to OAG is critical for designing specific interventions to better managing patients at risk of glaucomatous OND. Since only scarce studies on this topic exist in Spain and Portugal, a singular geographical area where the family and public culture, and lifestyle behaviors differs from other European countries [22], we decided to gather information on the conventional and other  non-ocular risk factors for the conversion of OHT to OAG in a Spanish and Portuguese population, mainly those in-volving metabolic, cardiovascular and respiratory diseases, lifestyle facts and habits (body mass index, coffee/tea intake, alcohol/tobacco use, psychotropic drug utilization) that could play an additional role in the development and progression of OAG.

  1. In method section, definitions of OHT, OAG should be mentioned. For example, diagnosing standard of OND sign, grading standard of OHT, diagnosing standard of mild glaucoma stage, and so on.

According to the Reviewer indications, we have added two new paragraphs (marked in yellow), to better clarify the requested data, in the Material and Methods section, as follows:

The OHT diagnosis was done as a condition in which the IOP is greater than 21 mmHg, with normal visual fields, normal optic disc, and retinal nerve fiver layer (RNFL), as well as a gonioscopy demonstrating an open anterior chamber angle. The OAG is a chronic progressive optic neuropathy with characteristic morphologycal changes at the optic nerve head (ONH) and RNFL in the absence of other ocular disease or congenital anomalies. Progressive retinal ganglion cells (RGCs) death and VF loss are associated with these changes.

 Classification of defects was done as: early (mean deviation (MD) values of −6dB or better), moderate (MD between −6 and −12 dB) or severe (MD of −12 dB or worse) glaucomatous loss. Therefore, glaucomatous damage staging into the appropriate categories was applied to the study participants to better managing the disease. Glaucoma OND includes typical optic nerve head alterations such as neuro-retinal rim thinning, splinter hemorrhages, peripapillary nerve fiber loss, asymmetry of cupping between patient's eyes and parapapillary atrophy among others. Data were registered as OPHTHAL in a Microsoft Excel spread sheath and conveniently reviewed by two independent ophthalmologists.

  1. In Methods section, the authors mainly collected 635 patients and parts of them suffered from OAG, did these patients had ever been treated either by medicine or surgery? Why had they not been treated or how did the author manage the influence of medicine on IOP?

Regarding this point, we have to explain the Reviewer that initially we collected 635 patients, half of these (approximately) suffering OAG, that were treated with hypotensive eye drops, laser or surgery, depending on the individual necessities. Also some of the OHT individuals (the other half of the initial sample size) were treated with hypotensive eye drops, whenever the clinical characteristics pointed to the necessity of doing it. However, the final sample size was 412 participants, as reflected at the end of page 4, that were classified as OHT individuals (n= 198; 48%) and OAG patients (n= 214; 52%). Major causes of the reduction in sample size were: volunteer withdrawn, clinical findings that strongly recomend to exclude the participant, personal causes or exceptional complications that impide to accomplish with the study course.  

Two new sentences have been enclosed in the Material and Methods section (page 3 ) and marked in yellow.

All patients diagnosed of OAG and some participants of the OHTG were treated with hypotensive eye drops, laser, or glaucoma surgery, depending on the individual necessities and the glaucoma stage.

It has to be considered that major causes of the reduction in sample size from the baseline selected participants to the end of study, were the following: volunteer withdrawn, clinical findings that strongly recommend excluding the participant, personal causes or exceptional complications that prevent to accomplish with the study course.

  1. Abbreviations should be defined carefully when they first appeared. For example, DEMP, QUEST, OPHTHAL, STAT. Please check thoroughly.   

Thanks indeed to the Reviewer for calling our attention to this matter. The whole text has been carefully revised. Furthermore, we have done a list of abbreviations that has been enclosed at the end of the ms (pages 12 and 13)

LIST OF ABREVIATIONS

^4 :                              Four Power

AH:                            Aqueous Humor

AIC:                            Akaike Information Criterion

ANOVA:                   Analysis of Variance

BCVA:                       Best Correct Visual Acuity

BMI:                           Body Mass Index

C:                                Cubic

CCT:                           Central Corneal Thickness

COPD:                       Chronic Obstructive Pulmonary Disease

DEMO:                     Register of socio-demographic data in the Microsoft Excel

            program

DXGLAUCOMA:    Glaucoma diagnosis

DXOHT:                    Ocular Hypertension diagnosis

EGPS:                         European Glaucoma Prevention Study

GS:                            Glaucoma Suspect

GWAS:                       Genome-wide association study

IOP:                            Intraocular Pressure

L:                                Linear

LE:                              Left Eye

MD:                            Mean Deviation

OAG:                         Open Angle Glaucoma

OAGG:                      Open Angle Glaucoma Group

OCPD:                       Obstructive Chronic Pulmonary Disease

OCT:                          Optica Cohenerence Tomography

OFTARED:               Ophthalmology network

OHT:                          Ocular Hypertension

OHTG:                       Ocular hypertension group

OHTS:                        Ocular Hypertension treatment Study

OND:                         Optic nerve degeneration

ONH:                         Optic Nerve Head

OPHTAL:                  Register of ophthalmologic data in the Microsoft Excel

                                   program

PARP-1:                     Poly Adenil di phosphate ribose polymerase-1

Q:                               Quadratic

QUEST:                     Register of Survey data in the Microsoft Excel program

RE:                             Right Eye

RGC:                          Retinal ganglion Cell

RICORS:                    Cooperative Research Network of Results Oriented to

                                   Health

RNLF:                        Retinal Nerve Fiber Layer

SARS-Cov2:              Sever acute respiratory syndrome Coronavirus 2

SBP:                            Systolic Blood Pressure

SD:                            Standard Deviation

SNRI:                         Serotonin Noradrenalin reuptake inhibitors

SSRI:                           Selective Serotonin reuptake inhibitors

STAT:                         Statistical Analysis

TM:                            Trabecular Meshwork

VF:                             Visual Function

  1. Please specify the main packages used in R. Also, please supply Python details as well as employed packages.

The authors thank the Reviewer for arising this important point of our work. We answer the Reviewer that all calculations were performed using R Statistics v4.1.1. R packages readxl, dplyr, ggplot2, ggcorrplot and DemoTools were used for data loading and wrangling, exploratory analysis, and data plotting. R packages caret, MASS, Epi, questionr and car were used for data modelling and model evaluation. A new sentence was enclosed at the end of the Material and Methods section to clarify these proceedings.

  1. According to the methods, the authors mainly tried to perform a multi-center study. The distribution of data from each center should be detailed in the results section, which would address the data bias concerns.

In agreement with the Reviewer comment, it has been done. A new sentence has been enclosed at the beginning of the Results section.

Universitary Hospital Dr. Peset of Valencia (22%); Universitary General Hospital of Valencia (20%); Clinic Hospital of Barcelona (20%); Universitary Hospital “Morales Meseguer” of Murcia (11%); Punta de Europa Hospital of Algeciras (11%); Universitary Jerez Hospital of Jerez de la Frontera (4%); Clinsborges of Porto, Portugal (8%); San Juan de Dios Hospital of Sevilla (4%).

  1. Y-axis as well as Y-scale should be shown in Figure 1.

We agree with the reviewer and have added the Y-axis and Y-scale in both figures 1 and 2.

  1. BCVA data should be treated after LogMAR transformation.

Yes, this has been done in agreement with the Reviewer Comments.

  1. The author should detail how CCT was measured (distance between which planes).

Once again. we are grateful to the Reviewer for this important comment. A new sentence has added on page 4 of the Material and Methods section, and marker in yellow for easy localization, to better clarify this issue.

The pachymeter was calibrated at the beginning of each session as the manufacturer´s recommendations. After instillation of anaesthetic eye drops, and undilated pupil, the ultrasonic pachymetry was performed, whose probe tip is approximately 1 mm in diameter.  The tip was placed perpendicularly on the plane of the central cornea. Three independent measurements were done in a random sequence of 3 minutes of each other. CCT values were considered as the mean + SD mm for three determinations for each participant. 

  1. As IOP data would be fluctuant even in continuous measuring, the author should mention how IOP data was collected (how many times were measured by how many physicians, and how the data were finally recorded).

We are grateful to the Reviewer for this comment. In fact a new sentence has been added in the Material and Methods section (page 4) and marked in yellow.

The IOP was measured three consecutive times at the ophthalmological visit, in any part of the time period between 09.00h and 12,00h (am). In healthy individuals the IOP can vary between 2-4 mm Hg in 24 h. Only the ophthalmologist collaborating with this study were responsibles for the IOP measurements by means of Goldman tonometer. IOP values were considered as the mean + SD mm Hg for three determinations for each participant.  All data were recorded in the corresponding data sheath named OPHTHAL.

  1. What does the bold margin mean in Figures? Details should be mentioned in the figure legend.

We thank the reviewer for pointing out this mistake. We have rebuilt the graphs removing those margins.

  1. In addition to Q12, what does the exact means in lines with dark background?

We might assume that the Reviewer refers to the lines with a dark background appearing in tables 2, 3, 4, and 5. These lines correspond to the variables with statistical significance. To avoid confusion, we have added the following sentence in the footer of these tables: "The row for each statistically significant variable is shown with a dark background".

  1. What does the exact meaning of legends in Y-axis. For example, Coffee1, COPD1, Alcohol1 and so on.

We thank the reviewer for this comment. That nomenclature is wrong. We have reconstructed the figure and correctly written the name of the variables.  The new figure 3 is now changed in the ms for the former one

  1. Since OAG is usually a bilateral disease, if the IOP was significantly different between two eyes, the author should mention why IOP were different. Otherwise, the data should be combined and treated as a whole.

Each patient and each eye had different IOPs.

However, as reported by Dinn et al., circadian variation of IOP in OAG patients is firmly concordant between both eyes of the same patient. It is plausible that the fellow eye IOP fluctuate assymetrically in short time periods. It has been recommended that using the uniocular trial should be aware of the limit of the IOP concordance. However, this has ben taken in part in the present work because we have evaluated and statistically processed data from the two eyes of each participant whenever it can be done.

One new sentence has been added in the Material and Methods section (Statistical processing) of the new version.

However, the authors would like to state that the “n” used in the study refers to participants, not to eyes. Each patient was labeled by their treating ophthalmologist as an OHT patient or an OAG patient and included in each subgroup. Final sample of participants was 412, that were clinically classified in 198 OHT individuals (48.06%) and 214 patients with OAG (51.94%).

Dinn RB, Zimmerman MB, Shuba LM, Doan AP, Maley MK, Greenlee EC, Alward WL, Kwon YH. Concordance of diurnal intraocular pressure between fellow eyes in primary open-angle glaucoma. Ophthalmology. 2007;114(5):915-20

  1. Before logistic regression analysis using all study variables, did the author ever check the collinearity among variables? Results should be detailed shown.

This is an important issue, and we thank the reviewer for his insightful comment. The analysis of collinearity among variables was done, by determining the variance inflation factor (VIF). However, we did not include this in the manuscript.

Values of this factor greater than 5 indicate high collinearity and the variable should be discarded. When calculating it for the first model, which includes all the variables, we observed high VIF values for height, weight and BMI, as expected and in correspondence with what is shown in the correlation figure. For the other two models, and after eliminating one of the three variables, the VIFs decrease to acceptable values.

We have added the VIF values in tables 3, 4 and 5, as additional columns.

Table 3. Logistic regression analysis using all study variables.

Variable

Coefficient

SD

Z-value

p*

OR

95% CI

GVIF

Df

GVIF^(1/(2*Df))

Intercept

-10.188

7.835

-1.300

0.193

-

Age (lineal)

-0.320

0.532

-0.603

0.546

0.726

0.252-2.086

1,418

4

1.045

Age (quadratic)

0.571

0.449

1.271

0.203

1.771

0.728-4.342

Age (cubic)

0.023

0.319

0.075

0.940

1.024

0.545-1.921

Age (4th power)

0.196

0.234

0.839

0.401

1.217

0.769-1.930

Gender

-0.050

0.307

-0.165

0.869

0.951

0.518-1.737

1.839

1

1.355

Thyroid dysfunction

0.252

0.350

0.719

0.472

0.777

0.386-1.536

1.137

1

1.066

Smoking

-0.501

0.329

-1.524

0.127

1.651

0.868-3.166

1.211

1

1.100

Alcohol drinking

0.017

0.326

0.054

0.957

0.983

0.514-1.858

1.139

1

1.067

Height (cm)

5.601

4.825

1.161

0.245

270.816

0.037-50,149,000

16.174

1

4.021

Weight (kg)

-0.115

0.060

-1.923

0.054

0.891

0.773-0.996

53.854

1

7.338

BMI (kg/m2)

0.339

0.157

2.157

0.031

1.405

1.050-2.040

38.328

1

6.191

Psychochemicals

0.208

0.252

0.828

0.407

0.812

0.493-1.326

1.127

1

1.061

Sleep apnoea

0.216

0.421

0.514

0.607

0.805

0.345-1.821

1.164

1

1.079

Asthma

1.365

0.465

2.934

0.003

0.255

0.097-0.609

1.106

1

1.052

COPD

-0.938

0.446

-2.103

0.035

2.555

1.077-6.279

1.139

1

1.067

Coffee

0.103

0.245

0.420

0.674

0.902

0.557-1.462

1.078

1

1.038

Tea

0.139

0.397

0.350

0.725

0.870

0.392-1.881

1.104

1

1.051

Migraine

-0.946

0.293

-3.226

0.001

2.577

1.460-4.628

1.193

1

1.092

Peripheric Vasospasm

0.225

0.269

0.837

0.402

0.798

0.468-1.349

1.144

1

1.070

SD: standard deviation; OR: odds ratio; CI: confident interval; Df: degrees of freedom; BMI: body mass index; COPD: chronic obstructive pulmonary disease. * Significance was set at 0.05. The row for each statistically significant variable is shown with a dark background.

Table 4. Logistic regression analysis with Akaike Information Criterion for iterative elimination of variables

Variable

Coefficient

SD

Z-value

p*

OR

95% CI

GVIF

Df

GVIF^(1/(2*Df))

Intercept

-1.328

0.693

-1.917

0.055

-

Smoking

-0.557

0.305

1.822

0.069

1.745

0.959-3.193

1,064360

1

1,03167824

Weight

-0.041

0.014

-2.869

0.004

0.959

0.932-0.987

BMI

0.139

0.044

3.140

0.001

1.149

1.055-1.257

Asthma

-1.382

0.450

-3.067

0.002

0.251

0.098-0.582

1,063217

1

1,03112414

COPD

0.797

0.428

1.860

0.063

2.219

0.969-5.274

1,048717

1

1,02406885

Migraine

0.864

0.272

3.173

0.001

2.372

1.398-4.076

1,056211

1

1,02772127

SD: standard deviation; OR: odds ratio; CI: confident interval; Df: degrees of freedom; BMI: body mass index; COPD: chronic obstructive pulmonary disease. * Significance was set at 0.05. The row for each statistically significant variable is shown with a dark background.

Table 5. Logistic regression analysis with “Akaike Information Criterion” for iterative elimination of study variables, by adding AGE and GENDER

Variable

Coefficient

SD

Z-value

p*

OR

95% CI

GVIF

Df

GVIF^(1/(2*Df))

Intercept

-1.332

0.754

-1.766

0.077

-

Age (lineal)

-0.299

0.514

-0.582

0.560

0.741

0.266-2.053

1,278755

4

1,031213

Age (quadratic)

0.500

0.438

1.142

0.253

1.650

0.692-3.948

Age (cubic)

0.031

0.314

0.099

0.921

1.031

0.555-1.915

Age (4th power)

0.169

0.230

0.735

0.462

1.184

0.754-1.862

Gender

-0.047

0.290

-0.163

0.870

0.953

0.537-1.683

1,660385

1

1,288559

Smoking

0.495

0.311

1.590

0.111

1.641

0.891-3.035

1,108708

1

1,052952

Weight

-0.046

0.017

-2.648

0.008

0.954

0.921-0.987

4,879696

1

2,209003

BMI

0.162

0.051

3.176

0.001

1.175

1.066-1.303

4,129089

1

2,032016

Asthma

-1.389

0.453

-3.063

0.002

0.249

0.096-0.581

1,082203

1

1,04029

COPD

0.883

0.440

2.005

0.044

2.418

1.031-5.881

1,115538

1

1,05619

Migraine

0.852

0.278

3.060

0.002

2.346

1.365-4.082

1,097906

1

1,04781

SD: standard deviation; OR: odds ratio; CI: confident interval; Df: degrees of freedom; BMI: body mass index; COPD: chronic obstructive pulmonary disease. * Significance was set at 0.05. The row for each statistically significant variable is shown with a dark background.

 17. Did the author ever check the cross-function among variables?

We are grateful to the Reviewer for this suggestion. We did not perform a cross-function analysis among variables. The reason is the same than in the previous question (Q16). Before the multivariate logistic regression, we analyzed the collinearity among variables by determining the variance inflation factor (VIF). And this analysis provided us with enough information to proceed with logistic regression.

  1. 95% CI of OR should be supplied in Table 4.

We agree with this Reviewer suggestion. We have added the CI in tables 3, 4 and 5, as can be seen above. Thank you so much for outstanding help.

  1. In last paragraph of results section, “HTO/OAG” should be “OHT/OAG”?

Yes please. Thank you again for your interest in our work.

  1. In the discussion section, since the author mentioned that tea consumption may be a potential risk factor in affecting IOP, positive or negative relations between tea consumption or tea related consumption dose should be mentioned. Because throwing out only a concept would be useless.

Thanks to the Reviewer for this suggestion. We enclosed some concepts regarding the tea consumption and OAG. However, data processing for tea consumption did not show statistical significance. Because of this, we only discussed superficially this matter.  As a result of the Reviewer suggestion, a comment about tea consumption and two new references have been included in the Discussion section (page 10) as follows:

Wu, C.M., Wu, A.M., Tseng, V.L., Y, F., Coleman, A.L. Frequency of a diagnosis of glaucoma in individuals who consume coffee, tea and/or soft drinks. Br J Ophthalmol. 2018;102(8):1127-1133

Gasiunas, K., Galgauskas, S. Green tea-a new perspective of glaucoma prevention. Int J Ophthalmol. 2022;15(5):747-752.

  1. The discussion is too long to extract a core. The author needs to reorganize the discussion section and highlight the key results.

Thanks so much to the Reviewer for this important comment. In fact, we have addressed these recommendations by shortening as much as possible the discussion section. Moreover, we have re-structured this section, as kindly requested. We also tried to emphasize the most significant results, by highlighting the most relevant data in the final sentence of the discussion.

  1. Why did the author mention thyroid gland after alcohol and tobacco part? Reorganizing the discussion was highly recommended.

The discusion has been shortened and extensively changed. The subsections of this part have also been re-structured (for instance, comorbidities as the main subsection including the endocrine-metabolic, vascular and respiratory disorders).  Changes have been marked in yellow.

  1. In the conclusion section, the author should highlight the key findings and meaning of affecting IOP. In the current version, the author only threw out a concept without clear indicators. For example, tea consumption and coffee consumption, the glaucomatous suspects should higher or decrease the frequency and consumption.

A new sentence has been done, as requested by the Reviewer in the conclusion section.

We would like to thank specially the Reviewer 1, for outstanding help provided through an exceptionally good review process. We hope that the revised version of the manuscript that has been much improved with the English editting (certificated), and the answers to their concerns, will cover all aspects required in this review

Many Thanks.

Reviewer 2 Report

This artical aims to study the impact of the systemic comorbidities on ocular hypertension and open-angle glaucoma in a population of Spain and Portugal. This topic is interesting and the statistic analysis is good. However, therere some disadvantages in this manuscript:

1. Hypertension blood pressure and diabetes mellitus were reported previously as the related systemic risk factors for open angle glaucoma, however, the study didnt contain these two important factors. Authors should added these two factors into this study.

2. Why advanced open angle glaucoma were excluded in this study? 

Author Response

ANSWERS TO THE REVIEWERS

The Reviewers have brought up some points on our work, and we really appreciate the time used by the Guest Editor and the Reviewers 1 and 2, and the opportunity to clarify our research. We are extremely thankful for the Reviewers constructive comments and suggestions that helped our writing skills.

REVIEWER 2

We would like to thank the Reviewer for positive criticism regarding our work. Thank you for your kind comments on the statistical processing.

We have addressed extensive changes in the English language and style throught the ms and illustrations. In addition, we sent THE MS TO ENGLISH EDITING (certificated). The new version has been received today and it was sent to MDPI submitting system. Moreover, we have added a list of abreviations for a better reading of the ms.

Regarding your questions:

  1. Hypertension blood pressure and diabetes mellitus were reported previously as the related systemic risk factors for open angle glaucoma, however, the study didn’t contain these two important factors. Authors should added these two factors into this study.

We thank the Reviewer for calling our attention to this important point of our work. In fact, some sentences regarding the studies dealing with HBP and DM in OAG were included in the former ms, with their corresponding references, among them Al-Shamiri et al., 2020; Tham and Chen 2017; Grzybowski et al., 2020; Mc Monnies 2017… and many other researchers. Recognized non-IOP risk factors are mainly older age, thinner CCT, and the degree of severity of glaucoma. Also, myopia and the family history of glaucoma has been considered important risk factors affecting glaucoma. It has also been described that low ocular perfusion pressure, cardiovascular/cerebrovascular disease, systemic hypertension/ hypotension, diabetes mellitus and/or hypercholesterolemia may play important roles in OAG course. However, we have done our best to gather information on “other” risk factors for OHT/OAG, and the above comorbidities have been considered outside the scope of this study, from the beginning, The suggestion arised by the Reviewer is now being processed for the authors and we will try to conduct new research including information on HBP and DM, as kindly addressed. Therefore, we would like to thank, again, the Reviewer for the suggestion, because we have included your point as a consideration for future study.

  1. Why advanced open angle glaucoma were excluded in this study? 

Thanks to the Reviewer for pointing out that the advanced OAG could be included in the study. Main purpose of this work was to identify non-ocular specific personal characteristics, systemic comorbidities, and lifestyle facts that may be involved in the interface between OHT and OAG. Therefore we want respectfully to state that the scope of our research was focused on OHT and early -to-moderate OAG stages in order to identify risk factors for the  conversion of OHT to OAG. Glaucoma Staging Codes (GSC) categorisation considers advanced glaucoma if there is evidence of glaucomatous optic disc and visual field loss in both upper and lower hemifields and/or a defect encroaching within 5° of fixation. Moreover, late presentation with advanced glaucoma has been widely related to higher medical resources use (clinical appointments, hypotensive eye drops use, glaucoma laser and surgery treatments, etc) and advanced glaucoma is also associated with a higher risk of blindness compared with early stage detection. In this context, it is really difficult to compare the personal characteristics and lifestyle facts of the OHT individuals and initial OAG patients with advanced glaucoma cases. A recent Sistematic Review on Advanced Glaucoma has been published by the Spanish Society of Ophthalmology, and we have also been part of this work. Therefore we realized that  it was really important to exclude advanced OAG patients from the present study to avoid a bias of fashion.

Finally, the authors of this collaborative work want to thank the Reviewer 2 for its positive criticism on our work and the useful revision of the manuscript.

Round 2

Reviewer 1 Report

Thank you very much for your detailed response. 

One more thing I am still concerned about is the interactions among variables (Q17 in last version). For example, among selected variables, whether patients with elder age, COPD, and a history of smoking own higher risks than those with COPD only. Because logistic regression is always calculated based on a hypothesis that variables were independent. But to our knowledge, COPD was more likely found in older and smoking patients, whether these patients own higher OAG risks. Interaction checking among features is still highly recommended.

Author Response

ANSWERS TO THE REVIEWERS ROUND 2.

  • REVIEWER 1, ROUND 2: RE-REVISED FORM

We extremely thank the Reviewer for helping us to clarify this question. First, following the recommendations of the Reviewer, we have verified the direct association between each of the three variables (SMOKING, COPD and AGE) and the diagnosis (DX: OHT/OAG). After this, we observed a noticeable association between COPD and DX (p=0.046), while the other two associations (SMOKING-DX and AGE-DX) lacked significance.

Always in agreement with the Reviewer suggestions, we have carefully checked whether the three variables (SMOKING, COPD and AGE_GROUP) are associated or not in our study population. We found a very significant association between SMOKING and COPD (p<0.001), and a significant association between AGE and SMOKING (p=0.032). As a result of the consultation with internal medicine colleagues, we can assume that SMOKING is a major cause of COPD. Therefore, we can conclude that elder smokers are at risk of COPD respect the global no smoking population of the same age, and that the above patients are at risk of OAG.

Regarding age, our data processing revealed no association between COPD and SMOKING for young individuals (under 55 years old). However, there is a clear association for those over 55 years old. This could indicate that the relationship between SMOKING and developing COPD would be stronger after years of tobacco consumption.

Taking these results into account, we have decided to eliminate the COPD variable from the logistic regression model and to build a new table 5, as follows:

Table 5. Logistic regression analysis with “Akaike Information Criterion” for iterative elimination of study variables, by adding AGE and GENDER

Variable

Coefficient

SD

Z-value

p*

OR

95% CI

GVIF

Df

GVIF^(1/(2*Df))

Intercept

-1.341

0.751

-1.785

0.074

-

Age (lineal)

-0.148

0.502

-0.296

0.767

0.862

0.317-2.336

1.241785

4

1.027438

Age (quadratic)

0.528

0.432

1.221

0.222

1.696

0.720-4.011

Age (cubic)

0.051

0.311

0.165

0.869

1.053

0.569-1.942

Age (4th power)

0.171

0.229

0.750

0.453

1.187

0.759-1.860

Gender

-0.103

0.288

-0.356

0.727

0.902

0.510-1.584

1,651071

1

1,284940

Smoking

0.624

0.305

2.049

0.041

1.866

1.029-3.410

1,062069

1

1,030567

Weight

-0.046

0.017

-2.594

0.009

0.955

0.922-0.988

4,825147

1

2,196622

BMI

0.162

0.051

3.188

0.001

1.175

1.066-1.301

4,092805

1

2,023068

Asthma

-1.399

0.453

-3.103

0.002

0.247

0.096-0.572

1,079335

1

1,038910

Migraine

0.849

0.278

3.062

0.002

2.338

1.364-4.059

1,099779

1

1,048704

SD: standard deviation; OR: odds ratio; CI: confident interval; Df: degrees of freedom; BMI: body mass index; * Significance was set at 0.05

The row for each statistically significant variable is shown with a dark background.

According to the new data, we changed the corresponding sentences in the abstract, material and methods, results, discussion, and conclusions.

In addition, we have paid for the Professional English Editting, in order to improve the language as much as possible.

Finally, the authors want to express their gratitude to the Reviewer for outstanding interest and exceptional help to improve our work.  THANK YOU SO MUCH

Reviewer 2 Report

Although the hypertension blood pressure and diabetes mellitus are not included, and authors want to explore some other risk factors for open angle glaucoma in this manuscript. 

Author Response

ANSWERS TO THE REVIEWERS, ROUND 2.

  • REVIEWER 2, ROUND 2: Although the hypertension blood pressure and diabetes mellitus are not included, and authors want to explore some other risk factors for open angle glaucoma in this manuscript.

Always in agreement with the kind suggestions from the Reviewer 1 we would like to express our gratitude to the Reviewer for helping us to better explain this issue. We have paid for the Professional English Editting, in order to improve the language as much as possible.

Following the Reviewer suggestions, we have emphasized the importance of the hypertension blood pressure and diabetes mellitus in relation to OAG was conveniently explained in the introduction (page 2, lines 94-100), and discussion (page 9, line 344; 348-351) sections. Also the major endpoint of our population study was specifically reflected in the introduction (page 2;  lines 103-111), material and methods (page 3; lines 147-159) and discussion  (page 9 , lines 357-452) sections.

The authors of the present study would like to give thanks to the Reviewer for outstanding interest, directed to improve our work.
